# Volume-preserving diffeomorphism as nonabelian higher-rank gauge symmetry

Yi-Hsien Du [1], Umang Mehta [1], Dung Xuan Nguyen[2,*], Dam Thanh Son[1]

**1** Kadanoff Center for Theoretical Physics, University of Chicago, Illinois 60637, USA

**2** Brown Theoretical Physics Center and Department of Physics, Brown University, 182 Hope Street, Providence, RI 02912, USA

\* dungmuop@gmail.com

December 14, 2021

## Abstract

We propose nonabelian higher-rank gauge theories in 2+1D and 3+1D. The gauge group is constructed from the volume-preserving diffeomorphisms of space. We show that the intriguing physics of the lowest Landau level (LLL) limit can be interpreted as the consequences of the symmetry. We derive the renowned Girvin-MacDonald-Platzman (GMP) algebra as well as the topological Wen-Zee term within our formalism. Using the gauge symmetry in 2+1D, we derive the LLL effective action of vortex crystal in rotating Bose gas as well as Wigner crystal of electron in an applied magnetic field. We show that the nonlinear sigma models of ferromagnets in 2+1D and 3+1D exhibit the higher-rank gauge symmetries that we introduce in this paper. We interpret the fractonic behavior of the excitations on the lowest Landau level and of skyrmions in ferromagnets as the consequence of the higher-rank gauge symmetry.

# 1  Introduction

Recently, considerable interest has been drawn to "higher-rank gauge theories," i.e., theories where the gauge potential is not a one-form, but a tensor of higher rank [1–14]. The physical motivation of the higher-rank gauge theories is the discovery of a new class of topological matter known as *fractons* [15–19], one feature of which is the existence of excitations with restricted mobility. The excitations either cannot move at all or can only move in lower-dimensional sub-spaces, while composites of elementary excitations can move freely. The restricted mobility of the fractonic excitations can be interpreted as the consequence of the higher-rank theories' conservation laws. In particular, the tensor Gauss's law leads to the conservation of not only the electric charge but also the electric dipole moment and, in some cases, higher moments of the charge distribution [1, 2, 19]. This has the physical effect of rendering the electric charge immobile but leaving the dipoles mobile or partially mobile. One representative example is the so-called "traceless scalar charge theory," in which the electric charges are immobile and the electric dipole can move only in the direction perpendicular to the dipole moment [2]. The higher-rank gauge theories have also been applied to describe defects in solids [20–24], supersolids [25], superfluid vortices [26, 27], smectics [5, 27, 28], and other systems.

All the higher-rank gauge symmetries in the physical models mentioned above are abelian.

There were attempts to generalize these symmetries to nonabelian symmetries [6, 29–31]; however, physical systems that realize those symmetries have not been explicitly proposed. In this paper, we propose nonabelian higher-rank symmetry theories for condensed matter systems of physical relevance. We reformulate the tensor gauge transformation in the traceless scalar charge theory in 2+1D [1, 2], showing that the gauge transformation is nothing but the volume-preserving diffeomorphism (VPD) in the linearized form. We then construct nonlinear higher-rank gauge theories with VPDs as the symmetry group. The nonabelian nature of the gauge symmetry has a nontrivial consequence: the charge density operators at different positions do not commute. Instead, they form the long-wavelength limit of the Girvin-MacDonald-Platzman (GMP) algebra [32–34], which reveals a connection to the lowest Landau level (LLL).

It is easy to notice that many features of the physics on the LLL bear a close resemblance to that of the field-theory models with higher rank symmetries [35]: for example, electric charges are pinned to one place by the large magnetic field, and neutral excitations (e.g., the composite fermion in the half-filled Landau level [36]) carry an electric dipole moment and can move in the direction perpendicular to the direction of motion. In this paper, we show that this resemblance is not accidental; in fact, the nonlinear higher-rank symmetry is realized as a symmetry of the problem of charged particles on the LLL. The relation between the dipole moment and momentum of an excitation in models with higher-rank symmetry was noticed in Refs. [13, 26].

We also will present several physical systems that enjoy the nonabelian higher-rank symmetry. In 2+1D systems, the symmetry originates from the lowest-Landau-level limit, where the volume-preserving nature of the diffeomorphisms comes from the restriction that the transformations should preserve the background magnetic field. In addition to the derivation of the GMP algebra, we draw a connection between the topological Wen-Zee term [37] with the Chern-Simons term in a higher-rank gauge theory.

Furthermore, we will use the gauge symmetry to derive the effective theories of the Wigner crystal of electrons in a strong magnetic field and the vortex crystal in a rotating Bose gas.

Finally, we find that the nonlinear sigma models describing ferromagnetism in 2+1D and 3+1D also exhibit the global higher-rank symmetry. The higher rank gauge symmetry provides a new interpretation of the conservation of multipole moments in ferromagnets [38]; it also explains the close resemblance between the behaviors of skyrmions in ferromagnets and charged particles in a magnetic field [39].

## 2 Review of the traceless scalar charge theory

For the paper to be self-contained, in this Section we will review the symmetric tensor gauge theory proposed by Pretko [1, 2]. We consider a higher-rank gauge theory called "traceless scalar charge theory," where the gauge potential is a symmetric rank-2 tensor $A_{ij}$: $A_{ij} = A_{ji}$.

Its conjugate momentum is the electric field $E_{ij}$. They satisfy the canonical commutation relation

$$[E_{ij}(\mathbf{x}),\, A_{kl}(\mathbf{y})] = i(\delta_{ik}\delta_{jl} + \delta_{il}\delta_{jk})\delta(\mathbf{x} - \mathbf{y}). \tag{1}$$

One imposes the Gauss law and the traceless constraint:

$$\partial_i\partial_j E_{ij} = \rho, \tag{2}$$

$$E \equiv E_{ii} = 0, \tag{3}$$

which lead to the conservation of the following charges:

$$\int d\mathbf{x}\, \rho(\mathbf{x}), \quad \int d\mathbf{x}\, \mathbf{x}\rho(\mathbf{x}), \quad \int d\mathbf{x}\, x^2\rho(\mathbf{x}). \tag{4}$$

The conservation of the quantities listed in in Eq. (4) imply that a charge cannot move, and a dipole can move only along the direction perpendicular to the dipole moment [1].

The constraints (2) and (3) generate the gauge transformations on the gauge potential

$$A_{ij} \rightarrow A_{ij} + \partial_i\partial_j\lambda, \tag{5}$$

$$A_{ij} \rightarrow A_{ij} + \delta_{ij}\mu. \tag{6}$$

For our purpose, it is convenient to use the second gauge transformation (6) to explicitly fix the gauge $A \equiv A_{ii} = 0$ and eliminate the trace of $A_{ij}$ from the set of dynamical degrees of freedom. Since the two constraints $E = 0$ and $A = 0$ do not commute according to the commutation relation (1), following Dirac one should modify the commutators, replacing them by the Dirac brackets [40], which in our case is

$$[O_1, O_2] \rightarrow [O_1, O_2]_{\mathrm{D}} = [O_1, O_2] + [O_1, E][E, A]^{-1}[A, O_2] - [O_1, A][E, A]^{-1}[E, O_2] \tag{7}$$

The new commutator is then

$$[E_{ij}(\mathbf{x}),\, A_{kl}(\mathbf{y})] = i\left(\delta_{ik}\delta_{jl} + \delta_{il}\delta_{jk} - \frac{2}{d}\delta_{ij}\delta_{kl}\right)\delta(\mathbf{x} - \mathbf{y}). \tag{8}$$

The Gauss constraint $\partial_i\partial_j E_{ij} = \rho$ generates now the gauge transformation

$$A_{ij} \rightarrow A_{ij} + \partial_i\partial_j\lambda - \frac{1}{d}\delta_{ij}\partial^2\lambda, \qquad \partial^2 \equiv \partial^k\partial_k. \tag{9}$$

To construct a gauge-invariant Lagrangian, we introduce the field strengths. The electric field

$$E_{ij} = \partial_i\partial_j A_0 - \frac{1}{d}\delta_{ij}\partial^2 A_0 - \partial_t A_{ij}, \tag{10}$$

is obviously gauge invariant. One notices that

$$\omega_i = -\partial_j A_{ij} \tag{11}$$

transforms like a $U(1)$ vector potential,

$$\omega_i \rightarrow \omega_i - \partial_i\tilde{\lambda}, \qquad \tilde{\lambda} = \left(1 - \frac{1}{d}\right)\partial^2\lambda, \tag{12}$$

using which one can define the magnetic field,

$$H_{ij} = \partial_i \omega_j - \partial_j \omega_i = -\partial_i \partial_k A_{jk} + \partial_j \partial_k A_{ik}, \tag{13}$$

that is manifestly gauge invariant. The simplest Lagrangian for the gauge field is then the "Maxwell theory,"

$$L = c_1 E_{ij}^2 - c_2 H_{ij} H_{ij}. \tag{14}$$

However, as noticed in Ref. [41], in (2+1)D, another possible term in the Lagrangian is the Chern-Simons term which, up to an overall coefficient, reads

$$L_{\text{CS}} = \varepsilon^{ij} (A_0 H_{ij} - A_{ik} \dot{A}_{jk}). \tag{15}$$

As for a Chern-Simons term, the Lagrangian density is gauge-invariant only up to a total derivative. This term is more relevant than the Maxwell term. The higher-rank Chern-Simons theory in 3 spatial dimensions was also discussed previously in Ref. [42].

Note that one does not have to require the theory to contain dynamical gauge fields in order to have the conserved quantities (4). A theory coupled to background gauge fields $(A_0, A_{ij})$ and is invariant under the gauge symmetry under which the gauge fields transform as

$$A_0 \to A_0 + \dot{\lambda}, \qquad A_{ij} + \partial_i \partial_j \lambda - \frac{1}{d} \delta_{ij} \partial^2 \lambda, \tag{16}$$

will have the conservation law

$$\partial_t \rho - \partial_i \partial_j J_{ij} = 0, \tag{17}$$

where $\rho$ and $J_{ij}$ are the operators that couple to $A_0$ and $A_{ij}$, respectively, and $J_{ii} = 0$. This is sufficient to derive the conservation of the quantities (4). In fact, in most of this paper, we will consider theories coupled to nondynamical background gauge fields.

## 3 Nonlinear higher-ranked symmetry

### 3.1 Traceless scalar charge theory in (2+1)D as a theory of linearized gravity

We now show that the theory presented in the previous section can be interpreted as a theory of linearized gravity, and the higher-rank symmetry is the linearized version of VPD. Instead of $A_{ij}$ we introduce an equivalent field $h_{ij}$ defined as

$$h_{ij} = -\ell^2 (\varepsilon_{ik} A_{jk} + \varepsilon_{jk} A_{ik}), \tag{18}$$

where $\ell$ is some constant of the dimension of length[1]. Note that $h_{ij}$ is also symmetric and traceless. The gauge transformation for $h_{ij}$ is inherited from (16)

$$h_{ij} \to h_{ij} - \ell^2 (\varepsilon_{ik} \partial_j \partial_k + \varepsilon_{jk} \partial_i \partial_k) \lambda. \tag{19}$$

---

[1] We assume $A_0$ has dimension 1 and $A_{ij}$ is of dimension 2, so $h_{ij}$ is dimensionless.

If we define

$$\xi^i = \ell^2 \varepsilon^{ik} \partial_k \lambda, \tag{20}$$

then the transformation law of $h_{ij}$ can be reformulated as

$$h_{ij} \rightarrow h_{ij} - \partial_i \xi_j - \partial_j \xi_i. \tag{21}$$

The transformation (21) is nothing but the transformation of the metric under the volume-preserving (or in 2D, area-preserving) diffeomorphism $x^i \rightarrow x^i + \xi^i$, since $\partial_i \xi^i = 0$ due to the definition (20). The connection between a higher-rank gauge theory and a linearized gravity theory was proposed previously in Ref. [43].

## 3.2   Nonlinear higher-rank symmetry

The fact that the gauge symmetry resembles the transformation law of a metric under VPDs allows one to devise a nonlinear version of the gauge symmetry. Namely, in our nonlinear theory, instead of a gauge field $h_{ij}$ (or $A_{ij}$ for that matter), the degree of freedom is the metric $g_{ij}$. The tracelessness of $h_{ij}$ translates into the statement that the metric is unimodular: $\det g = 1$. The linear theory is restored when one expands the metric around the flat metric: $g_{ij} = \delta_{ij} + h_{ij} + O(h^2)$.

Under an infinitesimal VPD $x^i \rightarrow x^i + \xi^i = x^i + \ell^2 \varepsilon^{ij} \partial_j \lambda$, the metric transforms as

$$\delta_\lambda g_{ij} = -\xi^k \partial_k g_{ij} - g_{kj} \partial_i \xi^k - g_{ik} \partial_j \xi^k = -\ell^2 \varepsilon^{kl} (\partial_k g_{ij} + g_{kj} \partial_i + g_{ik} \partial_j) \partial_l \lambda. \tag{22}$$

Now we need to write down the nonlinear version of the transformation laws for $A_0$. One notices that the VPDs do not commute: from Eq. (22) one reads

$$[\delta_\alpha, \delta_\beta] = \delta_{[\alpha, \beta]}, \tag{23}$$

with $[\alpha, \beta] = \ell^2 \varepsilon^{ij} \partial_i \alpha \partial_j \beta$. This means that our gauge symmetry is nonabelian; in this paper, we will use "nonlinear" and "nonabelian" interchangeably. The transformation of $A_0$ must satisfy the commutation relation (23). One can check that this can be accomplished by the following simple modification

$$\delta_\lambda A_0 = \partial_t \lambda - \xi^k \partial_k A_0 = \partial_t \lambda - \ell^2 \varepsilon^{kl} \partial_k A_0 \, \partial_l \lambda. \tag{24}$$

The transformation of $A_0$ in Eq. (24) was motivated by the symmetries of the lowest Landau level that will be discussed subsequently. Nonetheless, it is the unique nonlinear generalization of (16) given that the transformation is at most linear in $A_0$ and respects rotational invariance. Furthermore, the second term of (24) means that $A_0$ transforms as a scalar field under time-independent spatial diffeomorphism, which is expected. We leave the detailed discussion on the uniqueness of the nonlinear transformation (24) to Appendix A.

Equations (24) and (22) give the transformation laws of a nonlinear higher-rank symmetry, which we collect here for convenience:

$$\delta_\lambda A_0 = \partial_t \lambda - \ell^2 \varepsilon^{kl} \partial_k A_0 \, \partial_l \lambda, \tag{25}$$

$$\delta_\lambda g_{ij} = -\ell^2 \varepsilon^{kl} (\partial_k g_{ij} + g_{kj}\partial_i + g_{ik}\partial_j)\partial_l \lambda. \tag{26}$$

A nonlinear transformation similar to Eq. (26) was considered in Ref. [14] within a dynamical gauge model of the traceless scalar charge theory. One can derive the Ward identity from Eqs. (25) and (26). Let us define the charge density $\rho$ and the stress tensor $T^{ij}$ by varying the logarithm of the partition function

$$\delta \ln Z = \int d^3x \left( \rho \delta A_0 + \frac{1}{2} T^{ij} \delta h_{ij} \right). \tag{27}$$

The Ward identity is then

$$\dot{\rho} - \ell^2 \varepsilon^{kl} \partial_l \left[ \rho \partial_k A_0 + \tfrac{1}{2} T^{ij} \partial_k g_{ij} + \partial_i (T^{ij} g_{jk}) \right] = 0 \tag{28}$$

In the presence of the background field, only the total charge is conserved, but not the higher multipoles in (4).

Since the total charge is conserved, it is possible to introduce a vector potential $A_i$ so that the theory is invariant under the usual $U(1)$ gauge symmetry $A_\mu \to A_\mu + \partial_\mu \alpha$. In this case $A_0$ plays a double role: it is the temporal component of a U(1) gauge field $(A_0, A_i)$, and also as the scalar component of the gauge potential of a higher-spin symmetry, $(A_0, g_{ij})$. The two sets of gauge potentials share one scalar component, see Fig. 1. We will see an example when we consider ferromagnets (Sec. 6.4).

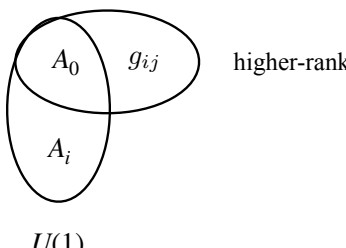

Figure 1: $A_0$ is shared by two sets of gauge potentials.

The nonabelian nature of the gauge symmetry, in some cases, allows us to derive the algebra satisfied by the charge density of the matter coupled to the gauge field. Imagine that the action $S_{\mathrm{m}}(\psi, A_0, g_{ij})$ describing the coupling of the matter fields $\psi$ with the gauge fields $(A_0, g_{ij})$ does not contain the time derivatives of any fields, $A_0$, $g_{ij}$, or $\psi$. In this case, if one promotes the gauge fields to dynamical fields by adding to the action a pure gauge action $S_{\mathrm{g}}$,

$$S = S_{\mathrm{g}}[A_0, g_{ij}] + S_{\mathrm{m}}[\psi, A_0, g_{ij}], \tag{29}$$

then upon quantization, the canonical commutation relations in the gauge sector are set by $S_{\mathrm{g}}$ and in the matter sector by $S_{\mathrm{m}}$. The left-hand side of the Gauss constraint,

$$\frac{\delta S_{\mathrm{g}}}{\delta A_0} + \frac{\delta S_{\mathrm{m}}}{\delta A_0} = 0, \tag{30}$$

is the generator that generates gauge transformations. In particular, in the matter sector, the commutator of the charge density $\rho(\mathbf{x})$ with any matter field $O(\mathbf{x})$ will give the change of $O$ under infinitesimal gauge transformation:

$$\left[ \int d\mathbf{y} \, \lambda(\mathbf{y})\rho(\mathbf{y}), \, O(\mathbf{x}) \right] = i\delta_\lambda O(\mathbf{x}). \tag{31}$$

But diffeomorphisms do not commute, so we conclude that the charge density at different points does not commute in our theory. We find

$$[\rho(\mathbf{x}), \, \rho(\mathbf{y})] = i\ell^2 \varepsilon^{ij} \partial_i \rho(\mathbf{x}) \partial_j \delta(\mathbf{x} - \mathbf{y}). \tag{32}$$

Here we recover the long-wavelength version of the Girvin–MacDonald–Platzman (GMP) algebra [32] (or the $w_\infty$ algebra), which suggests that the symmetry described here is related to the physics of the LLL.

## 4   Connection to quantum Hall effect

To establish the connection with the physics of the LLL, we recall the symmetry of the problem. A system of particles interacting with an electromagnetic field can also be put in curved space:

$$S = \int dt \, d\mathbf{x} \, \sqrt{g} \left( \frac{i}{2} \psi^\dagger \overleftrightarrow{\partial}_t \psi + A_0 \psi^\dagger \psi - \frac{g_{ij}}{2m} D_i \psi^\dagger D_j \psi + \cdots \right), \tag{33}$$

where $\cdots$ includes interaction terms. (Strictly speaking, the discussion here corresponds to the $g = 2$, $s = 1$ version of the LLL symmetry [44].) One can check that the classical action is invariant with respect to time-dependent spatial diffeomorphisms (i.e., all diffeomorphism transformations which preserve the time slices) [44, 45]:

$$\delta A_0 = -\xi^k \partial_k A_0 - A_k \dot{\xi}^k, \tag{34}$$

$$\delta A_i = -\xi^k \partial_k A_i - A_k \partial_i \xi^k - m g_{ik} \dot{\xi}^k, \tag{35}$$

$$\delta g_{ij} = -\xi^k \partial_k g_{ij} - g_{ik} \partial_j \xi^k - g_{ik} \partial_j \xi^k. \tag{36}$$

The LLL limit corresponds to taking $m \to 0$. The term proportional to $m$ disappears from the transformation law for $A_i$; now $A_\mu$ simply transforms like a one-form under spatial diffs:

$$\delta A_\mu = -\xi^k \partial_k A_\mu - A_\lambda \partial_\mu \xi^\lambda, \qquad \xi^\lambda = (0, \xi^i). \tag{37}$$

The metric $g_{ij}$ also transforms like a covariant tensor. The nontrivial feature of the states on the LLL that sets it apart from other states in a magnetic field is that, although a time-varying diffeomorphism generates the $g_{0i}$ components of the metric tensor from $g_{ij}$: $\delta g_{0i} = \cdots + g_{ij}\dot{\xi}^j$, the partition function of the theory does not depend at all on $g_{0i}$.

The fractional quantum Hall effect exists in a finite magnetic field $B = \partial_1 A_2 - \partial_2 A_1$. Suppose one is not interested in computing the electric current by varying the partition function with respect to $A_i$. In that case, one can assume that $A_i$ has some fixed value, for example, $A_i = -\frac{1}{2}B\varepsilon_{ij}x^j$, and only consider $A_0$ and $g_{ij}$ as external backgrounds. Then it is natural to ask if the partition function of the theory is symmetric under any gauge transformation that touches only $A_0$ and $g_{ij}$, and explore the Ward-Takahashi identities that follow. To keep $B$ unchanged, we need to restrict ourselves to VPDs. These correspond to $\xi^k = \ell^2 \varepsilon^{kl} \partial_l \lambda$, where we chose $\ell$ to be the magnetic length $\ell = 1/\sqrt{B}$. For VPDs, the change of the spatial components of gauge potential $A_i$,

$$\delta A_i = -\ell^2 \varepsilon^{kl} \partial_l \lambda \partial_k A_i - \ell^2 A_k \varepsilon^{kl} \partial_i \partial_l \lambda = -\ell^2 \varepsilon^{kl} \partial_l \lambda (\partial_k A_i - \partial_i A_k) - \ell^2 \partial_i (\varepsilon^{kl} A_k \partial_l \lambda)$$
$$= -\partial_i (\lambda + \ell^2 \varepsilon^{kl} A_k \partial_l \lambda), \quad (38)$$

can be compensated by a gauge transformation $A_\mu \to A_\mu + \partial_\mu \alpha$ with $\alpha = \lambda + \ell^2 \varepsilon^{kl} A_k \partial_l \lambda$. Under this combination of coordinates and gauge transformations, $A_0$ transforms as

$$\delta A_0 = -\ell^2 \varepsilon^{kl} \partial_l \lambda \partial_k A_0 - \ell^2 A_k \partial_t (\varepsilon^{kl} \partial_l \lambda) + \dot{\lambda} + \ell^2 \partial_t (\varepsilon^{kl} A_k \partial_l \lambda) = \dot{\lambda} - \ell^2 \varepsilon^{kl} \partial_k A_0 \partial_l \lambda. \quad (39)$$

We see that the transformation law for $A_0$ has exactly the form that we have postulated in Eq. (24). The metric, of course, transforms as in Eq. (22).

## 4.1 Higher-rank symmetry in the lowest Landau level limit

Within the context of the LLL, it is possible to give an intuitive interpretation of the higher-rank conservation law. We write down the current conservation

$$\frac{\partial \rho}{\partial t} + \boldsymbol{\nabla} \cdot \mathbf{j} = 0, \quad (40)$$

and the law of conservation of momentum,

$$\frac{\partial \pi_i}{\partial t} + \partial_j T_{ij} = E_i \rho + \varepsilon_{ik} j_k B, \quad (41)$$

where $\pi_i$ is the momentum density. In a Galilean-invariant theory with particles of mass $m$, the momentum density is proportional to the particle number flux $\pi_i = m j_i$, and vanishes in the LLL limit $m \to 0$. Now the conservation of momentum becomes simply the equation of balance of force, which in the absence of the electric field simply reads

$$\partial_j T_{ij} = \varepsilon_{ik} j_k B, \quad (42)$$

and can be solved to yield for the current

$$j_i = -\frac{1}{B} \varepsilon_{ij} \partial_k T_{jk}. \quad (43)$$

The equation for the conservation of charge is now

$$\frac{\partial \rho}{\partial t} - \frac{1}{2B}\partial_i\partial_j(\varepsilon_{ik}T_{kj} + \varepsilon_{jk}T_{ik}) = 0. \tag{44}$$

One notices that the conservation law (44) is in the same form as (17) in the earlier version of the symmetric tensor gauge theory of fracton. Thus, the conservation of charge has a "higher-rank" form due to the fact that, on the LLL, the current density is no longer independent but can be expressed through the derivative of the stress tensor. The same conservation law was derived previously in Ref. [46] using a LLL field theory formalism. The connection between volume-preserving diffeomorphism and quantum Hall physics was also noticed in Refs. [14, 26, 33, 34, 47–50].

Some comments are in order. We began with two independent Ward's identities, (40) and (41). The conservation law (44) can be considered as the linear combination of the charge conservation (40) and the massless limit of (41). One then recognizes that we end up with two independent conservation laws, one being (44) and the other being the original charge conservation. It is the same conclusion that we arrived at in Section 3.

## 4.2 The Wen-Zee term

One possible term in the effective action for the fractional quantum Hall is the Wen-Zee term [37]. To introduce this term, we need to define the Newton-Cartan geometry and the spin connection that comes with it. We only give the relevant formulas here; for details, see, e.g., Refs. [44, 51]. The Newton-Cartan geometry structure is given by a one-form $n_\mu$ (in the simplest version of the geometry $dn = 0$), a vector $v^\mu$, and a symmetric contravariant metric tensor $g^{\mu\nu}$ satisfying $n_\mu v^\mu = 1$, $g^{\mu\nu}n_\nu = 0$. In our case,

$$n_\mu = (1, \mathbf{0}), \qquad v^\mu = \begin{pmatrix} 1 \\ v^i \end{pmatrix}, \qquad g^{\mu\nu} = \begin{pmatrix} 0 & 0 \\ 0 & g^{ij} \end{pmatrix}. \tag{45}$$

where

$$v^i = \ell^2\varepsilon^{ij}\partial_j A_0, \tag{46}$$

and $g^{ij}$ is the inverse matrix of $g_{ij}$. One then defines the covariant metric tensor $g_{\mu\nu}$ so that $g_{\mu\nu}v^\nu = 0$ and $g_{\mu\nu}g^{\nu\lambda} = \delta_\mu^\lambda - n_\mu v^\lambda$,

$$g_{\mu\nu} = \begin{pmatrix} g_{ij}v^i v^j & -v_j \\ -v_i & g_{ij} \end{pmatrix}, \tag{47}$$

where $v_i \equiv g_{ij}v^j$, together with the Christoffel symbol which can be used to define covariant derivatives

$$\Gamma^\mu_{\nu\lambda} = v^\mu\partial_\nu n_\lambda + \frac{1}{2}g^{\mu\rho}(\partial_\nu g_{\rho\lambda} + \partial_\lambda g_{\rho\nu} - \partial_\rho g_{\nu\lambda}). \tag{48}$$

One further defines the vielbein $e_\mu^a$ so that

$$g^{\mu\nu} = e^{a\mu}e^{a\nu}, \tag{49}$$

and the spin connection is defined as

$$\omega_\mu = \frac{1}{2}\varepsilon^{ab}e^{a\nu}\nabla_\mu e^b_\nu, \tag{50}$$

which, in components, reads

$$\omega_0 = \frac{1}{2}(\varepsilon^{ab}e^{aj}\partial_0 e^b_j + \varepsilon^{ij}\partial_i v_j), \tag{51}$$

$$\omega_i = \frac{1}{2}(\varepsilon^{ab}e^{aj}\partial_i e^b_j - \varepsilon^{jk}\partial_j g_{ik}). \tag{52}$$

The spin connection transforms like the gauge potential under the local O(2) rotation of the vielbein: $e^a(x) \to e^a(x) + \alpha(x)\varepsilon^{ab}e^b(x)$. The Wen-Zee term [37] is given by

$$\frac{\kappa}{4\pi}\varepsilon^{\mu\nu\lambda}\omega_\mu\partial_\nu A_\lambda = \frac{\kappa}{4\pi}\left(\omega_0 B + A_0\varepsilon^{ij}\partial_i\omega_j - \varepsilon^{ij}\omega_i\partial_0 A_j\right) = \frac{\kappa}{4\pi}\left(\frac{\omega_0}{\ell^2} + \frac{1}{2}A_0 R\right), \tag{53}$$

where we have put $\varepsilon^{ij}\partial_i A_j = \ell^{-2}$ and $\partial_0 A_i = 0$. The coefficient $\kappa$ is related to the filling fraction $\nu$ and the Wen-Zee shift $\mathcal{S}$ of a fraction quantum Hall (FQH) system by the relation

$$\kappa = \nu\mathcal{S}. \tag{54}$$

Up to quadratic order and ignoring the total derivative terms, we can rewrite the Wen-Zee term as

$$\frac{\kappa}{8\pi}\left(A_0 R - \frac{1}{4\ell^2}\varepsilon^{ij}h_{ik}\dot{h}_{jk}\right), \tag{55}$$

which is exactly the Chern-Simons term (15).

A remark can be made here. We demonstrated that the Wen-Zee term in the FQH literature is nothing but the Chern-Simon term in the higher-rank gauge theory. One can think of this in the reversed order. The higher-rank gauge symmetry dictates the relationship between the Wen-Zee shift $\mathcal{S}$ and the Hall viscosity $\eta_H$, which enter two separated components of the Chern-Simons Lagrangian in the higher-rank gauge theory (55).

## 5  Generalization to (3+1) dimensions

### 5.1  Construction of the (3+1)D nonlinear higher-rank symmetry

This section will generalize the nonabelian higher-rank symmetry to (3+1) dimensions. To do that, we imagine a three-dimensional version of the LLL. Instead of a background magnetic field, we imagine a background Kalb-Ramond field. Concretely, we imagine a nonrelativistic theory living in background metric $g_{ij}$ and a Kalb-Ramond field $B_{\mu\nu} = -B_{\nu\mu}$. The field strength of the latter is

$$H_{\mu\nu\lambda} = \partial_\mu B_{\nu\lambda} + \partial_\nu B_{\lambda\mu} + \partial_\lambda B_{\mu\nu}, \tag{56}$$

and we assume that there is gauge symmetry with one-form gauge parameter $\alpha_\mu$ under which

$$\delta B_{\mu\nu} = \partial_\mu \alpha_\nu - \partial_\nu \alpha_\mu. \tag{57}$$

and $H_{\mu\nu\lambda}$ is invariant. Most crucially, we assume that our theory is invariant symmetry under time-dependent spatial diffeomorphisms,

$$\delta g_{ij} = -\xi^k \partial_k g_{ij} - g_{kj} \partial_i \xi^k - g_{ik} \partial_j \xi^k, \tag{58}$$

$$\delta B_{ij} = -\xi^k \partial_k B_{ij} - B_{kj} \partial_i \xi^k - B_{ik} \partial_j \xi^k, \tag{59}$$

$$\delta B_{i0} = -\xi^k \partial_k B_{i0} - B_{k0} \partial_i \xi^k - B_{ik} \dot{\xi}^k. \tag{60}$$

which is a 3D version of the $m \to 0$ (i.e., LLL) limit of the nonrelativistic diffeomorphism (34). We do not have a concrete example of a well-defined theory with the symmetry (58), (59) and (60). As the LLL can be thought of as the massless limit for particles in a magnetic field, one can imagine a theory of massless strings coupled to a Kalb-Ramond field. The details (or even the existence) of such a theory are not important for our further discussion.

Following our discussion of the LLL in (2+1)D, we assume that our system lives in a finite Kalb-Ramond field $H_{ijk} = \ell^{-3} \varepsilon_{ijk}$, and restrict ourselves to VPDs with $\partial_k \xi^k = 0$ or

$$\xi^k = \ell^3 \varepsilon^{klm} \partial_l \lambda_m. \tag{61}$$

The change of $B_{ij}$ under this VPD,

$$\delta B_{ij} = -\ell^3 \varepsilon^{klm} (\partial_k B_{ij} + B_{kj} \partial_i + B_{ik} \partial_j) \partial_l \lambda_m, \tag{62}$$

again can be compensated by a gauge transformation (57) with the gauge parameter

$$\alpha_i = \lambda_i - \ell^3 \varepsilon^{klm} B_{ik} \partial_l \lambda_m. \tag{63}$$

The combined VPD and gauge transformation changes only the $B_{0i}$ components of the Kalb-Ramond potential and the metric. This is a higher-rank symmetry that transforms our set of gauge fields $(B_{0i}, g_{ij})$ like

$$\delta B_{0i} = \dot{\lambda}_i - \ell^3 \varepsilon^{klm} (\partial_k B_{0i} + B_{0k} \partial_i) \partial_l \lambda_m, \tag{64a}$$

$$\delta g_{ij} = -\ell^3 \varepsilon^{klm} (\partial_k g_{ij} + g_{kj} \partial_i + g_{ik} \partial_j) \partial_l \lambda_m. \tag{64b}$$

In addition, we also inherit from the gauge symmetry (57) those transformations which leave $B_{ij}$ invariant. These correspond to gauge parameters with vanishing spatial components: $\alpha_\mu = (\alpha_0, \mathbf{0})$. Under these gauge transformations,

$$\delta B_{0i} = -\partial_i \alpha_0, \tag{65a}$$

$$\delta g_{ij} = 0. \tag{65b}$$

Equations (64) and (65) represent the full group of higher-rank symmetries. Though we have used an analogy with the LLL physics in 2D as a motivation, the resulting transformation laws

do not require any LLL-type microscopic physics. In fact, we will see that the higher-rank symmetry appears in the context of 3D ferromagnets.

As in 2D, it is possible to "complete" $B_{0i}$ by adding the spatial components $B_{ij}$ so that $B_{\mu\nu}$ form a set of Kalb-Ramond gauge potentials. This would make $B_{0i}$ the shared components of the two sets of gauge potentials: the Kalb-Ramond gauge potentials and the gauge potentials of the higher-rank symmetry of VPDs (see Fig. 2).

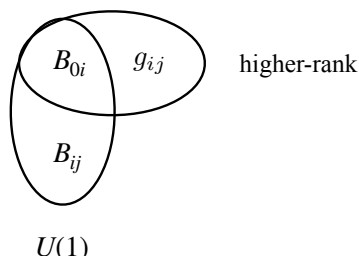

Figure 2: $B_{0i}$ is shared by two sets of gauge potentials.

## 5.2   Linearized higher-ranked symmetries and conservation laws

At the linearized level, the higher-rank gauge invariance is

$$\delta B_{0i} = \dot{\lambda}_i - \partial_i \alpha_0, \tag{66}$$

$$\delta h_{ij} = -\ell^3 (\varepsilon_{jkl}\partial_i + \varepsilon_{ikl}\partial_j)\partial_k \lambda_l. \tag{67}$$

As far as we know, this linear higher-rank symmetry has not been considered previously in the literature.

Let us now assume that the currents coupled to $B_{0i}$ and $h_{ij}$ are $J^i$ and $T^{ij}$,

$$\delta \ln Z = \int d^4 x \left( J^i \delta B_{0i} + \frac{1}{2} T^{ij}\delta h_{ij} \right). \tag{68}$$

Two conservation laws follow from Eqs. (66) and (67). First, the gauge invariance with gauge parameter $\alpha_0$ implies that the "current" $J^i$ is divergence-free:

$$\boldsymbol{\nabla} \cdot \mathbf{J} = 0. \tag{69}$$

On the other hand the VPD invariance, generated by $\lambda_i$, leads to

$$\frac{\partial J_i}{\partial t} - \ell^3 \varepsilon_{ijk}\partial_j \partial_l T_{kl} = 0. \tag{70}$$

This means the following quantities are conserved

$$\mathbf{I}_1 = \int d\mathbf{x}\, \mathbf{J}, \tag{71}$$

$$I_2 = \int d\mathbf{x}\, (\mathbf{x} \cdot \mathbf{J}), \quad \mathbf{I}_3 = \int d\mathbf{x}\, (\mathbf{x} \times \mathbf{J}), \quad I_4^{ij} = \int d\mathbf{x}\, x^{\{i} J^{j\}}, \tag{72}$$

$$\mathbf{I}_5 = \int d\mathbf{x}\, \mathbf{x}(\mathbf{x} \cdot \mathbf{J}), \qquad \mathbf{I}_6 = \int d\mathbf{x}\, \frac{x^2}{2}\mathbf{J}, \tag{73}$$

$$I_7^{ijk} = \int d\mathbf{x}\, x^{\{i} x^j J^{k\}}. \tag{74}$$

where the $\{ij \cdots\}$ denotes symmetrization over the indices $i, j, \cdots$ [2]. On the other hand,

$$I_8^{ij} = \int d\mathbf{x}\, x^{\{i}(\mathbf{x} \times \mathbf{J})^{j\}} \tag{77}$$

is not conserved; its time derivative is proportional to $\int d\mathbf{x}\, T^{ij}$.

Assume that the "current" $\mathbf{J}$ is nonzero only in a finite region of space, then because it is divergence-free, one can express it as the curl of a vector field: $\mathbf{J} = \boldsymbol{\nabla} \times \boldsymbol{\mu}$, where $\boldsymbol{\mu}$ vanishes at infinity. Then among the conserved quantities, only the following are nonzero: $\mathbf{I}_3$, $\mathbf{I}_5$, and $\mathbf{I}_6$, and the last two quantities are not independent:

$$\mathbf{I}_3 = \int d\mathbf{x}\, \boldsymbol{\mu}, \tag{78}$$

$$\mathbf{I}_5 = -\mathbf{I}_6 = \int d\mathbf{x}\, (\mathbf{x} \times \boldsymbol{\mu}). \tag{79}$$

One can think of $\boldsymbol{\mu}$ as the magnetic moment density and $\mathbf{J}$ as the magnetization current. Let us assume that there is a quasiparticle that carries a magnetic moment. Then the conservation of $\mathbf{I}_3$ means that the total magnetic moment does not change its value. The conservation of $\mathbf{I}_5$ implies that a particle can move only along the direction of its magnetic moment, but not along the two perpendicular directions.

Note that the (3+1)D higher-rank symmetry presented above, even in the linearized version, differs from that of the vector charge theory proposed in Ref. [2]. Our motivation was to generalize the area-preserving diffeomorphism of the LLL in (2+1)D to volume-preserving diffeomorphism of (3+1)D. We have generalized the charge density $\rho$, the one-form gauge potential $A_\mu$ and the constant magnetic field $B$ to the vector charge density $J^i$, the two-form gauge potential $B_{\mu\nu}$ and the constant Kalb-Ramond field strength $H_{ijk}$. We then arrive at a different gauge transformation rather than the one in Ref. [2]. One can define the electric field and magnetic field that are invariant under the gauge transformations and write down a

---

[2]Explicitly

$$A^{\{i}B^{j\}} = \frac{1}{2}\left(A^i B^j + A^j B^i\right), \tag{75}$$

and

$$A^{\{i}B^j C^{k\}} = \frac{1}{3}\left(A^i B^{\{j}C^{k\}} + B^i C^{\{j}A^{k\}} + C^i A^{\{j}B^{k\}}\right). \tag{76}$$

generalized Maxwell action. We will not do that here, as the aim of this paper is to investigate the conservation laws and their physical consequences[3].

The conservation law (70), though similar to the one of vector charge theory in Ref [2], is not the same because of the different gauge transformations. Recall that the gauge transformation of the symmetry tensor gauge in the vector charge version is [2,41]

$$\delta A_{ij} = \partial_i \lambda_j + \partial_j \lambda_i, \quad \delta \phi_i = \partial_t \lambda_i \tag{80}$$

which leads to a conservation law with only one spatial derivative acting on the current density $J^{ij}$

$$\partial_t \rho^i + \partial_j J^{ij} = 0 \tag{81}$$

instead of an equation with two spatial derivative. Furthermore, inherited from a system of vector matter field coupled with the Kalb-Ramond field, we have an *extra* "conservation law" $\vec{\nabla} \cdot \vec{J} = 0$, which does not have a counterpart in the vector charge theory proposed in Ref. [2].

We will show in Section 6.4 that the symmetry that has been proposed is realized in the nonlinear sigma model describing 3D ferromagnets.

# 6 Examples of theories with volume-preserving diffeomorphism invariance

For the abelian higher-rank symmetry, one of the simplest ways to couple a matter field to the gauge field $(A_0, h_{ij})$ is to introduce a Goldstone boson $\varphi$ which transforms under the gauge transformation as $\varphi \to \varphi + \lambda$ and write

$$\mathcal{L} = \frac{c_1}{2}(\partial_0 \varphi - A_0)^2 - \frac{c_2}{2}(\partial_i \partial_j \varphi - h_{ij})^2. \tag{82}$$

However, we were not able to find a nonlinear version of this transformation. For example, if one postulates

$$\delta_\lambda \varphi = \lambda - \ell^2 \varepsilon^{kl} \partial_k \varphi \partial_l \lambda, \tag{83}$$

then one can check by direct calculation that $[\delta_\alpha, \delta_\beta]\varphi \neq \delta_{[\alpha,\beta]}\varphi$, so such transformation law would be inconsistent. We have to devise other ways to couple the matter field to the gauge field.

## 6.1 Composite fermions

In the fractional quantum Hall effect at filling fractions $\nu = 1/2$, $\nu = 1/4$ etc., the quasiparticle is electrically neutral. One can consistently couple such a particle to the higher-rank gauge

---

[3]Since the gauge transformations (66) and (67) differ from the ones in Ref. [2], the definitions of the gauge-invariant electric field and magnetic field should be modified accordingly, namely $E^{ij} = (\varepsilon_{jkl}\partial_i + \varepsilon_{ikl}\partial_j)\partial_k B_{0l} + \partial_t h_{ij}$ and $H = \partial_i \partial_j h_{ij}$. We will not discuss them further.

field. For example, a Lagrangian for a nonrelativistic particle with dispersion relation $\omega = k^2/2m$ would be

$$L = \frac{i}{2}v^\mu \psi^\dagger \overset{\leftrightarrow}{\partial}_\mu \psi - \frac{1}{2m}g^{\mu\nu}\partial_\mu \psi^\dagger \partial_\nu \psi = \frac{i}{2}\psi^\dagger \overset{\leftrightarrow}{\partial}_0 \psi + \frac{i}{2}\ell^2 \varepsilon^{ij}\partial_j A_0 \, \psi^\dagger \overset{\leftrightarrow}{\partial}_i \psi - \frac{g^{ij}}{2m}\partial_i \psi^\dagger \partial_j \psi. \quad (84)$$

One can interpret the coupling of the electric field $\partial_i A_0$ with the particle momentum as a dipole moment, perpendicular to the direction of its motion. This is consistent with the constraints that follow from conservation laws. In fact, that is how a composite fermion in the fractional quantum Hall state at $\nu = 1/2$ or $\nu = 1/4$ should couple to the external potential: the composite fermion is neutral but possesses an electric dipole moment.

## 6.2 Crystal on the lowest Landau level

Another way to realize the higher-rank symmetry is through an effective theory of a solid. Such a solid may be realized as a Wigner crystal, which is expected to be the ground state of electrons on the LLL at small filling fractions. A solid is parametrized by a map from the physical coordinates $x^i$ to the coordinate system $X^a$ frozen into the solid: $X^a = X^a(x^i)$ [52]. In the ground state $X^a = x^i \delta_{ai}$, the displacement $u^a$ are defined as $X^a = x^a - u^a$. On the LLL the coordinates of a lattice site do not commute: $[x, y] = -i\ell^2$. The Lagrangian should thus contain the following term

$$S_{\text{Berry}} = \frac{n_0}{2\ell^2}\varepsilon^{ab}X^a\partial_t X^b, \quad (85)$$

where $n_0$ is the equilibrium particle number density. Under a VPD, $X^a$ transforms as

$$\delta_\lambda X^a = -\ell^2 \varepsilon^{ij}\partial_i X^a \partial_j \lambda, \quad (86)$$

and so

$$\delta_\lambda S_{\text{Berry}} = -\frac{n_0}{2}\varepsilon^{ij}\varepsilon^{ab}X^a\partial_i X^b\partial_j \dot{\lambda} = -\frac{n_0}{2}\dot{\lambda}\varepsilon^{ij}\varepsilon^{ab}\partial_i X^a\partial_j X^b. \quad (87)$$

This change of the action can be compensated by including a term proportional to $A_0$ into the Lagrangian. The full Lagrangian is then

$$\mathcal{L} = \frac{n_0}{2\ell^2}\varepsilon^{ab}X^a\partial_t X^b + \frac{n_0}{2}A_0\varepsilon^{ab}\varepsilon^{ij}\partial_i X^a\partial_j X^b - \varepsilon(O^{ab}), \quad (88)$$

with $O^{ab} = g^{ij}\partial_i X^a\partial_j X^b$ and $\varepsilon(O^{ab})$ is the energy associated with elastic deformations. The spectrum of this theory can be obtained by expanding the action to quadratic order over the displacement $u^a$. The presence of a term with a first time derivative implies that the dispersion relation of the lattice sound wave has the quadratic form $\omega \sim q^2$ rather than the linear form.

Thus, we have been able to construct the effective theory of a Wigner crystal on the LLL starting from the higher-rank symmetry.

### 6.3 Vortex crystal

Another type of matter is the "vortex crystal," which is realized, for example, in a rotating Bose gas [53]. The crystal is formed by the zeros of the condensate wave function. In this case, the lattice fields $X^a$ do not couple to $A_0$ directly, but through a dynamical gauge field $a_\mu$, which is the dual of the superfluid phonon. Under VPD $a_\mu$ transforms as a one-form,

$$\delta a_\mu = -\ell^2 \varepsilon^{kl} (\partial_k a_\mu - a_k \partial_\mu) \partial_l \lambda. \tag{89}$$

The field $a_\mu$ couples to the background $A_0$ through the following term

$$\frac{1}{2\pi} \int d^3x \left( \frac{a_0}{\ell^2} + A_0 b \right), \tag{90}$$

where $b = \varepsilon^{ij} \partial_i a_j$ is the emergent magnetic field. One can check directly that this term is invariant under VPDs. It is also obviously invariant under gauge transformations $a_\mu \to a_\mu + \partial_\mu \alpha$. In fact, (90) can be obtained from the Chern-Simons term $\frac{1}{2\pi} \varepsilon^{\mu\nu\lambda} a_\mu \partial_\nu A_\lambda$ by setting $A_i$ to be the static background with $\varepsilon^{ij} \partial_i A_j = \ell^{-2}$.

The Lagrangian of the vortex crystal is then determined by the symmetry and reads

$$\mathcal{L} = -\varepsilon^{\mu\nu\lambda} \varepsilon^{ab} a_\mu \partial_\nu X^a \partial_\lambda X^b - \varepsilon(O^{ab}) - \varepsilon_b(b) + \frac{1}{2\pi} \left( \frac{a_0}{\ell^2} + A_0 b \right), \tag{91}$$

where $\varepsilon(O^{ab})$ and $\varepsilon_b(b)$ represent the energies of the lattice and the condensate, respectively. Assuming $a_\mu$ transforms like a one-form under VPD, it can be checked that the Lagrangian above is invariant with respect to this symmetry. The eigenmode of this theory is again a Tkachenko mode with a quadratic dispersion relation [53].

Note that the above Lagrangian contains only leading-derivative terms and does not include next-to-leading terms considered in Ref. [53].

### 6.4 Ferromagnets

#### 6.4.1 Ferromagnets in 2+1 D

We now show that the ferromagnet in 2+1D secretly possesses a higher-rank symmetry similar to the models with particles on the LLL [4]. At the long-wavelength limit, a ferromagnet is described by a nonlinear sigma (NLS) model [55], written in terms of an $O(3)$ unit vector $n^a$, $n^a n^a = 1$, with the action

$$S = S_{\text{Berry}} + S_{\text{NLS}} = S_0 \int_0^1 d\sigma \int dt \, d\mathbf{x} \, \varepsilon^{abc} n^a \partial_t n^b \partial_\sigma n^c - \frac{J}{2} \int dt \, d\mathbf{x} \, \delta^{ij} \partial_i n^a \partial_j n^a. \tag{92}$$

The first term is a Wess-Zumino topological term in spin's action, which is induced by the Berry phase [55]. The second term is the energy term of the nonlinear sigma model and can be

---

[4]In fact, one can show that the dynamical equation of a single skyrmion in a 2-dimensional ferromagnet is the same as the equation of motion of a charged particle in a constant magnetic field [54].

made invariant under VPD by replacing $\delta^{ij}$ with $g^{ij}$. The first term is, however not invariant:

$$\delta_\lambda S_{\text{Berry}} = -S_0 \ell^2 \int_0^1 d\sigma \int dt\, d\mathbf{x}\, \varepsilon^{abc} \varepsilon^{ij} n^a \partial_i n^b \partial_\sigma n^c \partial_j \dot{\lambda}. \tag{93}$$

Integrating by parts, taking into account that $\epsilon^{abc} \partial_j n^a \partial_i n^b \partial_\sigma n^c = 0$ (this is because all the three O(3) vectors $\partial_i n^a$, $\partial_j n^a$, $\partial_\sigma n^a$ are perpendicular to $n^a$ and hence are linearly dependent) we find

$$\delta_\lambda S_{\text{Berry}} = S_0 \ell^2 \int_0^1 d\sigma \int dt\, d\mathbf{x}\, \varepsilon^{abc} \varepsilon^{ij} n^a \partial_i n^b \partial_j \partial_\sigma n^c \dot{\lambda}$$

$$= \frac{S_0}{2} \ell^2 \int_0^1 d\sigma \int dt\, d\mathbf{x}\, \varepsilon^{abc} \varepsilon^{ij} \partial_\sigma (n^a \partial_i n^b \partial_j n^c) \dot{\lambda} = \frac{S_0}{2} \ell^2 \int dt\, d\mathbf{x}\, \varepsilon^{abc} \varepsilon^{ij} n^a \partial_i n^b \partial_j n^c \dot{\lambda}. \tag{94}$$

We now choose

$$\ell^2 = \frac{1}{4\pi S_0}, \tag{95}$$

and add the following term to the action

$$S_{A_0} = -\frac{1}{8\pi} \int dt\, d\mathbf{x}\, A_0 \varepsilon^{abc} \varepsilon^{ij} n^a \partial_i n^b \partial_j n^c. \tag{96}$$

Then $\delta_\lambda(S_{\text{Berry}} + S_{A_0}) = 0$. Thus if we couple the ferromagnetic order parameter with the gauge fields $A_0$, $g_{ij}$ in the following way

$$S = S_{\text{Berry}} + S_{A_0} - \frac{J}{2} \int dt\, d\mathbf{x}\, g^{ij} \partial_i n^a \partial_j n^a, \tag{97}$$

then the action is invariant under VPD with $\ell^2$ defined in Eq. (95).

One can further introduce into the theory the vector gauge potential $A_i$, promoting Eq. (96) to

$$S_{A_\mu} = -\frac{1}{8\pi} \int dt\, d\mathbf{x}\, A_\mu \varepsilon^{abc} \varepsilon^{\mu\nu\lambda} n^a \partial_\nu n^b \partial_\lambda n^c. \tag{98}$$

with $A_i$ transforming as a one-form under VPDs. In this case, the potential $A_0$ is simultaneously the scalar component of the U(1) gauge field $(A_0, A_i)$ and the scalar component of the gauge potential of a higher-spin symmetry, $(A_0, g_{ij})$ (Fig. 1).

Now the scalar potential $A_0$ is coupled to the topological charge density $\rho(\mathbf{x})$. This means that a skyrmion behaves like a particle in an effective magnetic field with the magnitude

$$B_{\text{eff}} = -4\pi S_0 q, \tag{99}$$

where $q = \int d\mathbf{x}\, \rho(\mathbf{x})$ is the topological charge of the skyrmion. This fact can be derived by calculating the Berry phase associated with the motion of a skyrmion [39]. One also finds that the following quantities

$$\int d\mathbf{x}\, \rho, \quad \int d\mathbf{x}\, x^i \rho, \quad \int d\mathbf{x}\, x^2 \rho, \tag{100}$$

are conserved. This fact is again well known [38].[5]

It is instructive to rewrite the ferromagnet in the $\mathbb{CP}^1$ parametrization, where

$$n^a = z^\dagger \sigma^a z, \qquad z = \begin{pmatrix} z_1 \\ z_2 \end{pmatrix}, \qquad z^\dagger z = 1. \tag{101}$$

The action of the ferromagnet is then [54]

$$S = S_{Berry} + S_{NLS} = 2iS_0 \int dt\, d\mathbf{x}\, z^\dagger \partial_t z - 2J \int dt\, d\mathbf{x}\, D_i z^\dagger D_i z, \tag{102}$$

where $D_i z \equiv (\partial_i - ia_i)z$, and $a_i = -iz^\dagger \partial_i z$ is promoted to a dynamical field.

Now we couple the $\mathbb{CP}^1$ model to the external probes $(g_{ij}, A_0)$. We assume that under VPD, $z$ transforms as

$$\delta_\lambda z = -\ell^2 \varepsilon^{kl} \partial_k z \partial_l \lambda, \tag{103}$$

therefore,

$$\delta_\lambda S_{\text{Berry}} = -2iS_0 \ell^2 \int dt\, d\mathbf{x}\, \varepsilon^{ij} z^\dagger \partial_i z \partial_j \dot{\lambda} = -2iS_0 \ell^2 \int dt\, d\mathbf{x}\, \dot{\lambda} \varepsilon^{ij} \partial_i z^\dagger \partial_j z. \tag{104}$$

We now add the following term to the action:

$$S_{A_0} = 2iS_0 \ell^2 \int dt\, d\mathbf{x}\, A_0 \varepsilon^{ij} \partial_i z^\dagger \partial_j z. \tag{105}$$

Then $\delta_\lambda(S_{Berry} + S_{A_0}) = 0$, therefore the coupling of the ferromagnetic model with the gauge field $g_{ij}, A_0$ is

$$S = S_{\text{Berry}} + S_{A_0} - 2J \int dt\, d\mathbf{x}\, g^{ij} D_i z^\dagger D_j z, \tag{106}$$

and respects the higher-rank symmetry.

### 6.4.2    Ferromagnets in 3+1 D

For a ferromagnet in $(3+1)$D, the term $S_{A_0}$ that couples $A_0$ to the topological charge density is replaced by the coupling of $B_{0i}$ to the density of a one-form current,

$$S_{B_{0i}} = -\frac{1}{8\pi} \int dt\, d\mathbf{x}\, \varepsilon^{abc} \varepsilon^{ijk} B_{0i} n^a \partial_j n^b \partial_k n^c, \tag{107}$$

and this has the $(3+1)$D version of VPD invariance with $\ell^3 = (4\pi S_0)^{-1}$. Again one can promote (107) to

$$S_{B_{\mu\nu}} = -\frac{1}{8\pi} \int dt\, d\mathbf{x}\, \varepsilon^{abc} \varepsilon^{\mu\nu\lambda\rho} B_{\mu\nu} n^a \partial_\lambda n^b \partial_\rho n^c. \tag{108}$$

In this case, $B_{0i}$ are the shared components of a Kalb-Ramond gauge field and the set of gauge potentials of a higher-rank gauge symmetry $(B_{0i}, g_{ij})$ (see Fig. 2).

It was found in Ref. [38] that $\mathbf{I}_3$ and $\mathbf{I}_5 = -\mathbf{I}_6$ are conserved. We have given this fact a new interpretation in terms of a hidden higher-rank symmetry.

---

[5]In the presence of the Dyaloshinskii-Morya interaction, which breaks the higher-rank symmetry, only the first two quantities are conserved [56].

# 7   Conclusion

We have presented a nonlinear version of a higher-rank gauge symmetry. The symmetry is basically that of volume-preserving diffeomorphism. We show several examples of coupling of matter with the higher-rank gauge potential that respects the symmetry. Many examples are taken from the physics of the LLL, which we show to naturally have volume-preserving diffeomorphism invariance. We also show that the nonlinear sigma models of ferromagnetism also exhibit this symmetry if one couples a gauge potential of the higher-rank symmetry with the topological charge density.

We have shown that, under certain conditions, the charge densities satisfy the commutation relation of the VPD, i.e., the $w_\infty$ algebra. In this way, one can easily understand why this algebra is realized in the bimetric model of the FQH effect [57], without explicit calculations. One interesting question is whether the $w_\infty$ symmetry can be "upgraded" to the quantum $W_\infty$ symmetry [33, 34]. This question, relevant for the fractional quantum Hall effect, is deferred to future work.

## Acknowledgements

The authors thank Adrey Gromov and Sergej Moroz for discussions and comments on the earlier draft of this paper. This work is supported, in part, by the U.S. DOE grant No. DE-FG02-13ER41958, a Simons Investigator grant and by the Simons Collaboration on Ultra-Quantum Matter, which is a grant from the Simons Foundation (651440, DTS). DXN is supported by the Brown Theoretical Physics Center.

# A   The uniqueness of the nonlinear transformation (24)

In this Appendix, we will briefly argue that the nonlinear transformation (24) is the unique generalization of (16), given the following assumptions:

- The nonlinear transformation satisfies the area-preserving diffeomorphism algebra (23).

- In the background $(A_0, g_{ij})$, $A_0$ is a scalar, therefore the transformation $\delta_\lambda A_0$ should be a scalar.

- The transformation of $(A_0, g_{ij})$ is at most linear in $(A_0, g_{ij})$.

- The rotational symmetry is preserved.

Due to the algebra (23), the transformation $\delta_\lambda A_0$ has to be linear in $\lambda$. We consider the general transformation that is linear in $A_0$ with the form[6]

$$\delta_\lambda A_0 \sim (\partial_{i_1} \cdots \partial_{i_m}) A_0 (\partial_{j_1} \cdots \partial_{j_n}) \lambda, \tag{110}$$

with $\partial_k$ is the derivative in the spartial directions. By counting the number of derivative in the spartial directions, in order to satisfy the algebra (23), both $m$ and $n$ have to be 1. Furthermore, $\delta_\lambda A_0$ has to be a scalar and invariant under rotation. We need to contract all the spatial indices of the derivatives, $\varepsilon^{ij}$ is the only rotational invariant two indices tensor that helps us to satisfy the algebra (23)

$$\delta_\lambda A_0 = -\ell^2 \varepsilon^{ij} \partial_i A_0 \partial_j \lambda. \tag{111}$$

We also consider the term that is independent of $A_0$

$$\delta_\lambda A_0 = \alpha^{i_1 \cdots i_s} (\partial_t)^u (\partial_{i_1} \cdots \partial_{i_s}) \lambda - \ell^2 \varepsilon^{ij} \partial_i A_0 \partial_j \lambda, \tag{112}$$

with $\partial_t$ is the time derivative and $\alpha^{i_1 \cdots i_s}$ is a rotational symmetric tensor. By counting the derivatives, to satisfy (23) then $u + s = 1$. There is no rotational symmetric tensor with just one spartial index, thereofre $s = 0$ and $u = 1$. We then arrive at the transformation (24).

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
