# Peer review of "Volume-preserving diffeomorphism as nonabelian higher-rank gauge symmetry"

_SciPost Physics_

## Round 2 · Referee Report · Anonymous (Referee 1) · 2021-9-9

Report

In this paper the authors proof that linearised volume preserving diffeomorphisms (VPD) agree with the so-called higher rank gauge symmetry at the linear level. In addition they compare their construction with the lowest Landau level, Wigner crystals in a magnetic field, vortex crystals, and ferromagnets. The paper is scientifically sound, and establishes a dual view in the description of such relevant systems.

The manuscript has several typos the authors are probably already aware of, therefore I will not point them out here. I assume in a revise version they will correct them. On a more physical and technical level I have some questions/comments I would like the authors to address before I recommend the paper for publication.

  1. In section III.A, the higher rank field is interpreted as a linear perturbation of a spatial metric around the flat one, and connect gauge transformation with linearise VPD, however the non-linear transformation of the field A_0 in section III.B somehow is postulated without much justification. The embedding of the original symmetry into VPD does not seem to be unique. Therefore, some comments on the intuition of the authors to demand such transformation would be convenient. Are the authors claiming that in general higher rank symmetries are associated to physical systems with an underlying magnetic-like field?

  2. Eqs. 27 and 28 relate the fields A_0 and h_{ij} as the sources of a conserved charge \rho and the stress tensor T^{ij}, and claim that since \rho is conserved a usual gauge field A_i can be introduced . However, the continuity equation 28, relates the operator that couples to A_i with T^{ij}, it seems this two sources (h_{ij}, A_i) are not independent. In fact Eq. 43 supports this idea. Is not obvious the necessity of introducing A_i. I would appreciate if the authors could comment on that, it is a bit confusing to me.

  3. In section IV, and related with my previous comment, I wonder what is the physical interpretation of keeping a gauge field fixed under a gauge transformation. This section is dedicated to connect with quantum Hall, I would expect demanding transformations that keep B fixed, but not A_i.

  4. Section V discusses the generalisation of the proposal to 3+1 D, and uses a Kalb-Ramond field as a natural generalisation of the U(1) gauge field A_\mu. This model is intriguing since corresponds to a new class of vector charge higher-rank theory, the continuity equation contains two space derivatives on the current, therefore the cit is a three indices tensor. The usual vector charge systems I am aware of [2], contain a symmetric second rank current, and a continuity equation analogue to the momentum conservation one. Since the authors successfully relate the model with ferromagnetic system, I think more emphasis on their finding, and a comparison of the differences of these two classes vector charge systems would be convenient.

  5. What do the authors mean with the curly bracket notation in Eqs. 72, 74, 75?.

  6. I understand all subsections in VI basically show known results and systems, however for self consistency it is necessary some details on the definition of the objects. I believe this paper is of high interest to a broad audience, and some readers may be familiar with certain systems but not with others. I will enumerate a list a variables that are not properly introduced in the manuscript.

  7. The quantity b in equation 86 -The acronym NLS in Eq. 88?
  8. In Eqs. 89 and 93 I think there is a typo and it should be either J or f^2 in the last terms
  9. In subsection VI.D when discussing the 3+1D ferromagnets, the authors start with the set of fields n^a, B_{\mu\nu} and g_{ij}, then they define the fields z^\dagger and z however they end up with (z^\dagger,z,g^{ij}, a_i), unfortunately I didn't find the definition of a_i. In equation 104 should S_{A_0} be S_{B_{\mu\nu}}?

To conclude, I find the paper interesting and with potential impact. I think the results presented here should be published. However, given the questions and comments above I do not recommend it for publication in the present form.

  • validity: high
  • significance: high
  • originality: high
  • clarity: good
  • formatting: -
  • grammar: good

Author:  Dung Nguyen  on 2021-10-07  [id 1818]

(in reply to Report 1 on 2021-09-09)

In this paper the authors proof that linearised volume preserving diffeomorphisms (VPD) agree with the so-called higher rank gauge symmetry at the linear level. In addition they compare their construction with the lowest Landau level, Wigner crystals in a magnetic field, vortex crystals, and ferromagnets. The paper is scientifically sound, and establishes a dual view in the description of such relevant systems. The manuscript has several typos the authors are probably already aware of, therefore I will not point them out here. I assume in a revise version they will correct them. On a more physical and technical level I have some questions/comments I would like the authors to address before I recommend the paper for publication.

Reply: We thank the referee for the fruitful comments, we fixed typos in the new version of our manuscript.

Q:1. In section III.A, the higher rank field is interpreted as a linear perturbation of a spatial metric around the flat one, and connect gauge transformation with linearise VPD, however the non-linear transformation of the field A_0 in section III.B somehow is postulated without much justification. The embedding of the original symmetry into VPD does not seem to be unique. Therefore, some comments on the intuition of the authors to demand such transformation would be convenient. Are the authors claiming that in general higher rank symmetries are associated to physical systems with an underlying magnetic-like field?

Reply: The transformation was motivated by symmetries of the lowest Landau level and the diff transformation of a scalar field. Nonetheless, the non-linear transformation of $A_0$ in equation (24) is the unique transformation that is the generalization of the linear transformation (16) given the following assumptions: - The non-linear transformation satisfies the area-preserving diff algebra (Eq. (23)). - In the background $(A_0,g_{ij})$, $A_0$ is a scalar, therefore the transformation $\delta_{\lambda} A_0$ should be a scalar. - The transformation of $A_0$ is at most linear in $A_0$.
- Rotational symmetry is preserved. We added a comment under equation (24) to emphasize these points. We also added an Appendix to sketch the argument on the uniqueness of the non-linear transformation (24).

Q: 2. Eqs. 27 and 28 relate the fields A_0 and h_{ij} as the sources of a conserved charge \rho and the stress tensor T^{ij}, and claim that since \rho is conserved a usual gauge field A_i can be introduced . However, the continuity equation 28, relates the operator that couples to A_i with T^{ij}, it seems this two sources (h_{ij}, A_i) are not independent. In fact Eq. 43 supports this idea. Is not obvious the necessity of introducing A_i. I would appreciate if the authors could comment on that, it is a bit confusing to me.

Reply: In fact, we have two independent conservation laws (and two Ward’s identities consequently). One is the charge conservation as the usual U(1) in equation (40), and one is for momentum conservation which is equation (41). In the lowest Landau level limit, one can think of equation (41) as the force balance equation from which one can derive current density (or $\vec{\nabla}\vec{j}$) in terms of derivative of stress tensor $T^{ij}$ as in equation (43) as the referee noticed. Then the conservation law in equation (44) is the linear combination of (40) and (41). However, since we begin with two Ward’s identities, we should end up with two conservation laws after manipulations. The second one is the original charge conservation, equation (40). From (44) and (40), one can again relate the derivative of stress tensor $T^{ij}$ and current $j^i$. In summary, the conservation law (28) and the conservation of charge are independent. This is also true in the Lowest Landau level limit that we discussed in Section 4. We added a few comments at the end of Section 4.1 to clarify this point.

Q: 3. In section IV, and related with my previous comment, I wonder what is the physical interpretation of keeping a gauge field fixed under a gauge transformation. This section is dedicated to connect with quantum Hall, I would expect demanding transformations that keep B fixed, but not A_i.

Reply: The answer to this question is partly addressed above. One can indeed consider the transformation that keeps $B$ fixed instead of $A_i$ (The transformation of $A_i$ can be cancelled by a U(1) gauge transformation). Under this transformation, the conservation law should be modified in which it should include charge density $\rho$, current density $j^i$ and stress tensor $T^{ij}$. However, one can eliminate $j^i$ in the new conservation law using the charge conservation law, so the final conservation law includes only charge density and stress tensor. This is exactly the consequence of the gauge transformation that keeps $A_i$ unchanged instead of $B$.

Q: 4. Section V discusses the generalisation of the proposal to 3+1 D, and uses a Kalb-Ramond field as a natural generalisation of the U(1) gauge field A_\mu. This model is intriguing since corresponds to a new class of vector charge higher-rank theory, the continuity equation contains two space derivatives on the current, therefore the cit is a three indices tensor. The usual vector charge systems I am aware of [2], contain a symmetric second rank current, and a continuity equation analogue to the momentum conservation one. Since the authors successfully relate the model with ferromagnetic system, I think more emphasis on their finding, and a comparison of the differences of these two classes vector charge systems would be convenient.

Reply: In fact, the conservation law (70) looks like the conservation law of a vector charge theory. However, since our gauge transformation is distinct from the gauge transformation that generates the vector charge version of symmetric tensor gauge theory in Ref[2], our conservation law differs from one in Ref [2]. Moreover, the definitions of gauge-invariant electric field and magnetic field should be different. We added some comments at the end of section 5 in the revised version to distinguish the differences.

Q: 5. What do the authors mean with the curly bracket notation in Eqs. 72, 74, 75?.

Reply: The curly bracket means symmetrizing over the indices. We added the definition under equation 74. We also added footnote [2] to define the explicit definition in the case of the curly brackets with 2 and 3 indices.

Q: 6. I understand all subsections in VI basically show known results and systems, however for self consistency it is necessary some details on the definition of the objects. I believe this paper is of high interest to a broad audience, and some readers may be familiar with certain systems but not with others. I will enumerate a list a variables that are not properly introduced in the manuscript.

  • The quantity b in equation 86

In the revised manuscript, we defined $b$ under equation 86 (equation 93 in the new version).

-The acronym NLS in Eq. 88?

We added the explanation for NLS and the citation to Fradkin’s book for more details above equation 88 (equation 95 in the new version).

  • In Eqs. 89 and 93, I think there is a typo, and it should be either J or f^2 in the last terms

Thank you for noticing this typo. We changed the coefficient in equation 93 (equation 100 in the new version) to $J$ for consistency.

  • In subsection VI.D when discussing the 3+1D ferromagnets, the authors start with the set of fields n^a, B_{\mu\nu} and g_{ij}, then they define the fields z^\dagger and z however they end up with (z^\dagger,z,g^{ij}, a_i), unfortunately, I didn't find the definition of a_i. In equation 104 should S_{A_0} be S_{B_{\mu\nu}}?

Thank you for the comments. This is the extra section devoted to formulating Ferromagnet in 2+1D using $\mathbb{CP}^1$ parametrization. Therefore it should be $S_{A_0}$ in equation (104) instead of $S_{B_{\mu\nu}}$. To avoid confusion, we separate this section VI.D into two subsections, now 6.4.1 and 6.4.2.

We added the definition of new emergent gauge field $a_i$

We transferred the discussion on $\mathbb{CP}^1$ parametrization to subsection 6.4.1 (Ferromagnets in 2+1D).

Comment: To conclude, I find the paper interesting and with potential impact. I think the results presented here should be published. However, given the questions and comments above I do not recommend it for publication in the present form.

Reply: We thank the referee for expressing interest in our manuscript. We hope that we addressed the issues/comments satisfactorily that the referee raised.

---

## Round 3 · Referee Report · Anonymous (Referee 2) · 2021-11-1

Strengths

  1. The paper is clearly written, and derivations are easy to follow.

  2. Many examples are given where VPD can be found and lead to higher rank gauge symmetry upon linearization.

  3. Provides a unified way of understanding some known fracton-like properties in conventional condensed matter systems.

Weaknesses

  1. Many annoying typos are not corrected during the resubmission. Some of them can be easily fixed using a spellchecker or simply by carefully reading the manuscript.

  2. Some examples are very brief and do not provide a physical context. For example, in the conclusion section, there is a sentence, "We also show that the nonlinear sigma models of ferromagnetism also exhibits this symmetry, if one couples a gauge potential of the higher-rank symmetry with the topological charge density." While this is indeed shown in section 6, it remains mysterious what physical situation requires the presence of the gauge potential of higher rank. (also exhibits should be changed to exhibit)

Report

The central observation of the paper is that the higher-rank gauge symmetries (HRGS) can occur as a linearization of the symmetry under volume-preserving diffeomorphisms (VPD). Therefore, symmetry under VPD can be thought of as a nonlinear version of HRGS.

The authors give a few examples of known systems where symmetry under VPD can be realized by appropriate coupling to background fields. As the linearization of VPD produces HRGS, one can derive from the latter some of the fractonic properties in those examples. As a result, many known fractonic properties can be understood as emerging from the underlying symmetry under VPD. However, it is not clear under what conditions the HRGS can be promoted to VPD. The authors provide an example of the abelian HRGS (see eq. 85), which they could not lift to VPD.

All examples considered in the paper are essentially nonlinear sigma models with the metric background on a target space.

I find the paper interesting and worth publishing in SciPost Physics. The manuscript is clearly written except for a few points listed below.

Requested changes

  1. Multiple typos and grammar inconsistencies should be corrected. Examples of trivial typos are: "notice noticed", "that manifests the gauge invariant." (that is manifestly gauge invariant), "Up to quadratic order and ignore the total" (ignoring), and many others.

  2. In the generally phrased citation "The connection between volume-preserving diffeomorphism and quantum Hall physics was also noticed in Refs. [14] and [26]." it would be very relevant to refer to much earlier works (current Refs 33, 34, of the manuscript).

  • validity: high
  • significance: high
  • originality: high
  • clarity: high
  • formatting: excellent
  • grammar: acceptable

Author:  Dung Nguyen  on 2021-12-13  [id 2027]

(in reply to Report 1 on 2021-11-01)

Comment: Some examples are very brief and do not provide a physical context. For example, in the conclusion section, there is a sentence, "We also show that the nonlinear sigma models of ferromagnetism also exhibit this symmetry, if one couples a gauge potential of the higher-rank symmetry with the topological charge density." While this is indeed shown in section 6, it remains mysterious what physical situation requires the presence of the gauge potential of higher rank. (also exhibits should be changed to exhibit)

Reply: We did not consider the physical gauge fields that skyrmions couple to. In fact, in the manuscript, the background fields were used as a formal device to provide an alternative interpretation of the emergent symmetries (from which the conserved quantities follow).

In principle, one can think of a physical situation where the higher-rank gauge potential can be realized. The variation of metric $h_{ij}$ can be realized through the local deformation of the lattice (which induces the spatial dependent magnetic coupling). In addition, the source of skyrmion density (scalar potential $A_0$) is related to the applied spin-polarized current in a complicated way (One can see more detail in Chapter 7 of Ref. [54]). However, the topic is a little bit complicated, thus we do not discuss further the physical realization of the gauge potentials in our manuscript.

Comment: Multiple typos and grammar inconsistencies should be corrected.

Reply: We thank the referee for the comment. Indeed, in the revised manuscript, we have tried to carefully fix all the typos.

Comment: In the generally phrased citation "The connection between volume-preserving diffeomorphism and quantum Hall physics was also noticed in Refs. [14] and [26]." it would be very relevant to refer to much earlier works (current Refs 33, 34, of the manuscript).

Reply: We cited more references that studied the connection between volume-preserving diffeomorphism and quantum Hall physics, including Refs. 33, 34 and the new Refs. 47-50.

---

## Round 3 · Referee Report · Anonymous (Referee 1) · 2021-11-2

Report

The authors have satisfactorily addressed the concerns and comments of my previous report. The only point I still find weak, is the argument of treating the gauge field $A_i$ and the metric $h_{ij}$ as independent fields. Actually, in the quantum Hall example (LLL) the momentum conservation is more a constraint than a conservation equation, this fact somehow seems to relate the fields $A_i, h_{ij}$. However, this is a minor aspect and I think the content of the paper is innovative and scientifically sound, therefore, I believe it satisfies the publication criteria of SciPost, and I recommend it for publication.
  • validity: -
  • significance: high
  • originality: high
  • clarity: ok
  • formatting: good
  • grammar: reasonable

Author:  Dung Nguyen  on 2021-12-13  [id 2026]

(in reply to Report 2 on 2021-11-02)

We would like to thank the referee for the recommendation for publication and previous feedback that helped us improve our manuscript significantly.

---

## Round 3 · Author Response

To the editors of SciPost Physics:

We thank the referee for the careful reading of our manuscript and comments, which have helped us improve our work.

We have edited the SciPost Physics manuscript in line with these comments. We believe that after these revisions, we have answered all of these referees’ queries, and our work is now appropriate for publication in SciPost Physics.

Sincerely
Yi-Hsien Du, Umang Mehta, Dung X. Nguyen and Dam Thanh Son

---

## Round 3 · List of Changes

-We changed the format of the manuscript to SciPost Physics format.

-We fixed typos.

  • We added comments at the end of section 4.1 to discuss the two independent conservation laws. One is the consequence of the volume-preserving diffeomorphism (combining with a corresponding $U(1)$ gauge transformation). One is the charge conservation as the consequence of $U(1)$ gauge transformation only.

-We added some comments at the end of section 5 in the revised version to distinguish the differences between 3+1D higher-rank symmetry in this paper with the vector charge theory proposed in Ref[2].

  • We added the definition of curly brackets under equation 74 and footnote [2] to define the explicit definition in the case of the curly brackets with 2 and 3 indices.

  • Explained the acronym NLS in Eq. 88 (equation 95 in the new version) and add the reference for more details of the non-linear sigma model.

  • Added the definition of the emergent gauge field $a_i$ in section VI.D (Section 6.4 in the new version)

  • We split section VI.D to 6.4.1 and 6.4.2 to discuss Ferromagnets in 2+1D and 3+1D separately.
  • Moved the discussion of the $\mathbb{CP}^1$ parametrization to subsection 6.4.1 (Ferromagnets in 2+1D)
  • We added Appendix A to discuss the uniqueness of the non-linear transformation of $A_0$ in Eq. (24).

---

## Editorial Decision

resubmitted